# Is household contact investigation a missing link for tuberculosis care in Chhattisgarh, India?– Operational research using programmatic data

Chetanya Malik[1*◔], Vishnu Gupta[2◔], Kalpita Shringarpure[3], Himanshu Abhay Gupte[4], Hemant Deepak Shewade[5], Vikash Ranjan Keshri[6], Narayan Tripathi[7], Khemraj Sonwani[8], Yogeshwar Kalkonde[1], Yogesh Jain[1]

**1** Sangwari- People's Association For Equity And Health, Surguja, Chhattisgarh, India, **2** State Health Resource Center, Raipur, Chhattisgarh, India, **3** Department of Community Medicine, Medical College, Baroda, Vadodara, Gujarat, India, **4** Narotam Sekhsaria Foundation, Mumbai, Maharashtra, India, **5** Division of Health Systems Research, ICMR-National Institute of Epidemiology, Chennai, Tamil Nadu, India, **6** Jindal School of Public Health and Human Development, O P Jindal Global University, Sonipat, India, **7** School of Liberal Arts, Indian Institute of Technology, Jodhpur, Rajasthan, India, **8** Department of Health, Directorate of Health Services, Raipur, Chhattisgarh, India

◔ These authors have contributed equally.
* drchetanyamalik86@gmail.com

## Abstract

Household contact (HHC) investigation helps in early identification of people with tuberculosis (TB) and initiation of TB preventive treatment (TPT) among those at high risk of developing TB. This cross-sectional study uses National TB Elimination Program data of all people notified with bacteriologically confirmed pulmonary TB and their HHCs from October to December 2023, from Chhattisgarh, a central Indian state, to assess coverage of HHC investigation, proportions identified with TB and put on TPT (all age groups and age < 5 years). Sociodemographic, clinical, and health system-related factors were used to identify predictors of HHC investigation not done, as determined through modified Poisson regression. Of the 4,221 people notified with TB, an HHC investigation was conducted for 3,177 (75%) cases. Among a total of 11670 contacts screened, TB was diagnosed in 0.9%(n = 109) for all age groups and 0.7%(n = 9) for children<5 years. TPT was initiated in 66% (n = 7740) for all age groups and 73% (n = 903) for children<5 years. Women (adjusted prevalence risk aPR 1.10; 95%CI:1.01-1.19), those notified from non-tribal districts (aPR 1.14; 95%CI:1.01-1.29), current facility being tertiary care (aPR 1.50; 95%CI:1.12-2.00) and private (aPR 1.42; 95%CI:1.08-1.86) facility, diagnosed with test other than sputum microscopy (aPR NAAT 3.19; 95%CI:2.39-4.28; LPA 8.88 95%CI:6.15-12.82; culture 9.69; 95%CI:5.99-15.68) and for whom diabetes (aPR 1.40; 95%CI:1.16-1.70) and HIV screening (aPR 1.55, 95% CI:1.17-2.05) was missing predicted higher risk of HHC investigation not done. The study highlights the need to improve HHC investigation, as well as the low yield of TB and TPT initiation. Predictors of HHC investigation not done suggest a need to decentralize it to the primary level and improve

**Data availability statement:** All anonymized data has been made available as a figshare link https://figshare.com/s/2739d023cecddd25fadb.

**Funding:** The modular training under this SORT IT course was funded by the ICMR-National Institute of Epidemiology (ICMR-NIE), Chennai, India. ICMR-NIE did not provide any specific funding for the operational/ implementation research (that resulted in this manuscript) conducted through this SORT IT course. The research was conducted in routine operational settings utilizing existing health resources and workforce.

**Competing interests:** The authors have declared that no competing interests exist.

data-based program monitoring. A statewide capacity-building initiative for improving the investigation of HHC is the way forward.

## Introduction

Tuberculosis (TB) is the leading cause of death among infectious diseases globally, and India continues to be one of the high-burden countries. The estimated detection gap of TB between notified and expected cases globally is 2.4 million [1]. India reported 2.5 million cases in 2023 and missed 0.3 million cases [2].

Contact investigation for those living with pulmonary TB (index) cases is part of the strategy to improve TB outcomes, as per the END TB strategy [3]. This involves screening contacts for TB disease, early detection of TB disease, and linking them to treatment. If TB disease is ruled out, eligible contacts are initiated on TB preventive treatment (TPT) [4]. India launched and expanded the TPT coverage under the National TB Elimination Program (NTEP) to all household contacts in 2021. It prioritized bacteriologically confirmed pulmonary TB for the same [5]. TPT initiation among HHCs in India in 2023 was 27%. This is against the global and national targets of 90% [2].

Studies have shown the effectiveness of contact investigation in improving TB case notification and TPT coverage globally and in India, conducted in project settings in clinics or part of the TB program in few districts [6–10].

The Government of India mandates improving HHC investigation through decentralized service at health and wellness centres (HWCs), which provide primary health care services (to a population of 5000) [11]. The World Health Organization (WHO) recommends decentralized, family-centered, and integrated care models to deliver services across the entire spectrum of the TB care cascade [4]. Studies in high TB burden countries have shown the benefits of decentralized models of care in improving TB care for those with TB disease and infection [12,13].

This operational research in the central Indian state of Chhattisgarh is the first attempt to assess the state-level coverage of investigation of HHCs among adults with bacteriologically confirmed pulmonary TB in India. It reports the numbers and proportions of HHCs of those adults detected with TB or initiated on TPT -for all ages and for children <5 years. It also investigated the predictors of not undergoing HHC investigation at the index-TB level. The findings of this operational research will be used to guide a proposed statewide implementation project involving HWCs to improve HHC investigation, TB case detection, and TPT coverage.

## Methods

### Study design

This was a cross-sectional analytical study involving routinely collected program data.

### Study Setting

Chhattisgarh is an agrarian central Indian state with a population of 32 million. Eighty percent of the population resides in rural areas, and one-third of the population is

comprised of scheduled tribes (socially marginalised indigenous groups of people as per the Indian constitution). The tribal communities have poorer health and socio-economic indicators, and a higher prevalence of TB compared to the non-tribal population [14]. One-sixth of the population is poor as per the multidimensional poverty index [15].

Chhattisgarh is a state with a high prevalence of TB and a low prevalence of HIV and diabetes. Pulmonary TB case notification in the state (129 per 100 000),) is low when compared to the reported prevalence (373 per 100 000),) and is among the lowest in the country, indicating a need to strengthen the health system to improve the notification [2,16]. Chhattisgarh has 3500 HWCs and primary health centers (PHCs) at the primary care level, which have 778 designated microscopy centres for diagnosing TB, 150 community health centers(CHCs) which function as TB units, 33 district hospitals at the secondary level, and 11 medical colleges at the tertiary level in the public health system [2]. TB microscopy is conducted at the PHCs and above, nucleic acid-based amplification tests (NAAT) are conducted at the secondary care level, and culture, line probe assay, and other tests are conducted at the tertiary care level. All health facilities where a medical officer can diagnose and treat TB are referred to as peripheral health institutes (PHI). 7098 registered private providers include outpatient clinics, hospitals, and two private medical colleges [2].

## Specific setting

*Ni-kshay* is a web-enabled application for monitoring of TB program in India. The notification register in *Ni-kshay* captures data for all notified pulmonary TB. This includes sociodemographic, clinical, laboratory, and diagnosing health facility-related characteristics, as well as comorbidities (diabetes and HIV). HHC is a person who shares the same enclosed living space as those with index TB disease for one or more nights or for frequent or extended daytime periods during the three months to initiation of current TB treatment. The contact tracing register in *Ni-kshay* records aggregate numbers for HHC investigation (HHCs identified, contacts screened for TB, eligible for TPT, initiated on TPT, diagnosed with TB). TPT register in *Ni-kshay* records individual-level data for contacts initiated on TPT, including outcomes (this feature was started from January 2023) [2,17]. The program requires an HHC investigation to be conducted by program staff in the public health sector. Those with symptoms of TB are referred to PHIs for confirmation of diagnosis for TB. Those notified from TB units located in tribal areas are provided a financial incentive for their transport to PHIs [18]. Contacts < 5 years are to be started on TPT after ruling out TB, and those>5 years are to be put on TPT after testing for TB infection [5].

## Study population

All adults notified with pulmonary TB during October to December 2023, residing and taking treatment in Chhattisgarh, were included.

## Data extraction and analysis

Data were extracted from the notification register, contact tracing register, and TPT register of *Ni-kshay* in October 2024. All registers were downloaded for those on TB treatment whose current facility was in the state of Chhattisgarh (using the current filter in *Ni-kshay*), irrespective of the location of their diagnosing facility between October to December 2023 in Microsoft Excel format. Data were filtered for adults (age > 18 years) and those who were bacteriologically confirmed, and coded as per the codebook (see S1 File Annexure). The datasets from the notification register, contract tracing register, and TPT register were merged using a unique episode identifier.

Districts where more than 50% population belonged to scheduled tribes were labelled 'tribal' and those with a population less than 50% as 'non-tribal'. Those with current PHI at HWC and PHC were labelled 'primary health facilities', CHC and district hospital were labelled 'secondary health facilities and medical colleges were labelled as 'tertiary health facilities' in the public sector.

Key output variables were households of index TB undergoing contact investigation, TB diagnosis, and TPT initiation among all age groups and age < 5 years separately. After merging the databases, if no numbers (aggregate) were reported

in the contact tracing register, it was recorded as not undergoing HHC investigation. There were discrepancies in the numbers initiated on TPT based on aggregate information in the contact tracing register and individual-level data in the TPT register in *Ni-kshay* (significantly underreported in the TPT register). For TPT initiation, we therefore relied on the former. Due to the same reasons, we were unable to explore predictors of TPT completion. Individual-level variables of HHCs who were diagnosed with TB disease were similarly not available.

Variables related to the baseline characteristics of the index TB disease included age, gender, district (current), test used for diagnosis, bacteriological confirmation, history of previous treatment, HIV status, diabetes status, type of PHI, and type of diagnosing facility. Index TB level predictors for not undergoing HHC investigation were identified using modified Poisson regression with robust variance estimates. Unadjusted and adjusted prevalence ratios (and 95% confidence intervals) were calculated for baseline variables. Age, gender, and other variables with a p-value <0.20 on univariate analyses were considered for adjusted multivariate analysis. Analyses were done using EpiData Analysis software (v 2.2.2.183 EpiData Association Odense, Denmark) and STATA (v16.1, copyright 1985–2019, Stata Corp. LP College Station, TX, USA).

### Ethics

The ethics approval was obtained from the Institutional Human Ethics Committee of ICMR-National Institute of Epidemiology (ICMR-NIE), Chennai, India (NIE/IHEC/A/202408–04) on 11-09-2024. Since the study involved secondary data analysis of program data, a waiver for consent was sought from the ethics committee and approved for this study.

## Results

Of the 4221 households having an adult with bacteriologically confirmed pulmonary TB (index) between October to December 2023, contact investigation was conducted for 75% (n = 3177). Aggregate contact tracing register data showed a total of 11670 contacts being screened, of which TB was diagnosed in 0.9% (n = 109), and TPT was initiated for 66% (n = 7740) contacts. Of the identified contacts, 1230 were below the age of 5 years; of whom nine (0.7%) were diagnosed with TB and 73% (n = 903) were initiated on TPT. This was a secondary data analysis of the aggregate numbers of HHCs documented per index TB disease, rather than the total number of HHCs (**Fig 1**).

Of 4221 adults with bacteriologically confirmed pulmonary TB, 32% were women, and the mean age (standard deviation) was 44 (16.4) years. Twenty-five percent of adults (n = 1069) were from tribal districts, 20% were diagnosed in private healthcare facilities, 51% (n = 2140) were diagnosed in secondary healthcare facilities, and 11% (n = 453) were diagnosed in primary healthcare facilities. Confirmation of diagnosis was based on NAAT testing for 52% (n = 2178), and microscopy was conducted in 31% (n = 1315). Almost 10% (n = 390) of the notified adults had a previous history of TB. Diabetes status was documented as 'yes' in 10% (n = 429) of adults and missing in 8% (n = 358). HIV was confirmed in 1.4% (n = 61), and data were missing for 3.3% (n = 141) (**Table 1**).

Index case-related predictors of not undergoing HHC investigation were studied. On adjusted analysis, women (aPR 1.10; 95%CI:1.01-1.19), those notified from predominantly non-tribal districts (aPR 1.14; 95%CI:1.01-1.29), notified from private healthcare facilities (aPR 1.42; 95%CI:1.08-1.86) and public tertiary healthcare facilities (aPR 1.50; 95%CI:1.12-2.00), those with drug-resistant TB (aPR 3.38; 95%CI:2.57-4.44), bacteriological confirmation based on any test other than sputum microscopy (NAAT aPR 3.19; 95%CI:2.39-4.4.28: LPA aPR 8.88; 95%CI:6.15-12.82: culture aPR 9.69; 95%CI:5.99-15.68) and those with missing previous treatment status (aPR 2.24; 95%CI:1.86-2.68) or missing diabetes (aPR 1.40; 95%CI:1.16-1.70) or HIV status (aPR 1.55, 95%CI:1.17-2.05) were less likely to undergo HHC investigation. (**Table 1**)

## Discussion

### Key findings

This is the first study to utilize state-level program data to evaluate the coverage and yield of HHC investigation and index case-related predictors for not undergoing HHC investigation in Chhattisgarh, a state with a high TB burden among

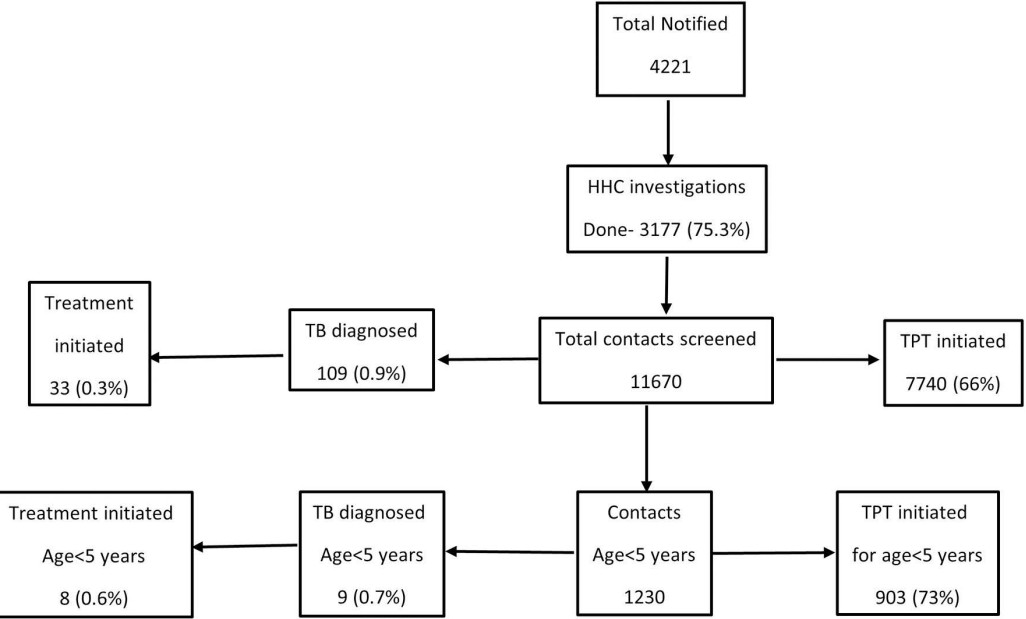

**Fig 1. TB case detection and TPT initiation among HHCs (overall and children <5 years) of those with adult notified bacteriologically con-firmed pulmonary TB between October- December 23 in Chhattisgarh, India.**

notified individuals with TB. The study shows sub-optimal coverage and yield of HHC investigations in diagnosing TB in all age groups and children<5 years. TPT was initiated among six of the ten contacts identified. Index case-level predictors for not undergoing HHC investigation were identified.

## Strengths and limitations

This operational research was conducted using routine program data, thus reflecting the real-world setting. This study focused on proximal factors affecting TPT coverage, while recent studies have focused on acceptance of TPT, newer regimens, and expansion to different contact groups. As this study utilized available secondary data from the state's TB program, the aggregate number of household contacts entered into the database was used for the study. This may not be all the contacts of those notified with TB, but only those for whom an HHC investigation was done, which was 75% of those notified. Secondly, even for those whose HHC investigation was done, not all contacts may be added to the portal. Thus, we could not determine the proportion of HHC screened out of the total contacts. Lack of individual-level data for those diagnosed with TB and those initiated on TPT through contact investigation meant their outcomes were not evaluated. The study looked at factors recorded at the time of notification only. The errors inherent in routinely collected data (secondary data) in operational research cannot be ruled out. Limitations notwithstanding, this study has many policy and practice-relevant findings.

## Relevance to policy and practice

Our study yielded 0.9% of TB among HHCs screened for TB in all age groups and 0.7% in children <5 years of age. This is lower than reported elsewhere. Recent meta-analysis from low- and middle-income settings reported a yield of 4.8% for all TB and 6.8% in children<5 years [19]. Similar meta-analysis from 2013 reported a yield of 3.1% in all age groups and 10% for children <5 years in lower-middle-income countries [20]. A clinic-based study done in Chennai, India, yielded 5.3% of TB disease [10]. A study done in a district of Chhattisgarh reported a yield of 9% [9]. However, in both these studies, contact investigation was completed in almost 100% of contacts.

**Table 1. Risk factors associated with not undergoing HHC investigation among adult bacteriologically confirmed pulmonary TB (index) cases notified during October- December 2023 (N = 4221) in Chhattisgarh, India.**

| Patient characteristics | N (%)ᶜ | HHC investigation not done (n) (%)ʳ | PR (95% CI) | aPR (95% CI) |
|---|---|---|---|---|
| Total | 4221 | 1044 (24.7) | | |
| Age | | | | |
| Age 18–44 y | 2218 (52.5) | 551 (24.8) | Ref (0.94-1.03) | |
| Age 45–59 y | 1113 (26.4) | 262 (23.5) | 1.02 (0.98-1.06) | 0.96 (0.88-1.05) |
| Age > 60 y | 890 (21.1) | 231 (26) | 0.99 (0.94-1.03) | 0.98 (0.88-1.09) |
| Gender | | | | |
| Male | 2879 (68.2) | 657 (22.8) | Ref | |
| Female | 1342 (31.8) | 387 (28.8) | 0.92 (0.89-0.96) | **1.10 (1.01-1.19)** |
| District | | | | |
| Predominantly Non-tribal | 3152 (74.7) | 884 (28.1) | 1.89 (1.62-2.20) | **1.14 (1.01-1.29)** |
| Predominantly Tribal | 1069 (25.3) | 159 (14.9) | Ref | Ref |
| Clinical characteristics | 4221 | 1044 | | |
| New | 3440 (81.6) | 747 (21.7) | Ref | Ref |
| Previously treated | 390 (9.2) | 49 (12.6) | 1.12 (1.07-1.16) | 0.88 (0.74-1.04) |
| Missing | 313 (7.4) | 170 (54.3) | 2.50 (2.22-2.82) | **2.24 (1.86-2.68)** |
| Resistant | 78 (1.8) | 78 (100) | 4.61 (4.32-4.91) | **3.38 (2.57-4.44)** |
| Basis of diagnosis | | | | |
| ZN staining | 1315 (31.2) | 4 8(3.7) | Ref | Ref |
| CBNAAT/Truenat | 2178 (51.6) | 282 (12.9) | 3.55 (2.63-4.78) | 3.19 (2.39-4.28) |
| LPA | 112 (2.7) | 101 (90.2) | 24.71 (18.59-32.83) | **8.88 (6.15-12.82)** |
| Culture | 11 (0.3) | 8 (72.7) | 19.92 (12.63-31.44) | **9.69 (5.99-15.68)** |
| Others | 605 (14.2) | 605 (100) | 27.40 (20.75-6.16) | **22.68 (17.15-29.99)** |
| HIV status | | | | |
| Non-reactive | 4019 (95.3) | 936 (23.3) | Ref | Ref |
| Reactive | 61 (1.4) | 14 (23) | 0.99 (0.62-1.57) | 1.00 (0.74-1.35) |
| Missing*** | 141 (3.3) | 94 (66.7) | 2.86 (2.51-3.26) | **1.55 (1.17-2.05)** |
| DM | | | | |
| Negative | 3434 (81.3) | 773 (22.5) | Ref | Ref |
| Positive | 429 (10.2) | 96 (22.4) | 0.99 (0.82-1.20) | 0.94 (0.84-1.05) |
| Missing*** | 358 (8.5) | 175 (48.9) | 2.17 (1.92-2.46) | **1.40 (1.16-1.70)** |
| Diagnosing health facility | | | | |
| Primary (HWC/PHC) | 453 (10.7) | 32 (7.1) | Ref | Ref |
| Secondary (CHC/DH) | 2140 (50.8) | 270 (12.6) | 1.79 (1.26-2.54) | 0.97 (0.74-1.26) |
| Tertiary (Medical college) | 382 (9) | 174 (45.5) | 6.45 (4.54-9.16) | **1.50 (1.12-2.00)** |
| Private** | 864 (20.5) | 509 (58.9) | 8.34 (5.94-11.70) | **1.42 (1.08-1.86)** |
| Others *** | 382 (9) | 59 (15.4) | 2.19 (1.45-3.29) | 1.05 (0.78-1.40) |

HHC- household contact, TB- Tuberculosis, ZN staining- Ziehl Neelsen staining, CBNAAT- Cartridge based nucleic acid amplification test, Truenat- a rapid molecular test based on polymerase chain reaction technology to detect TB, LPA- Line probe assay, DSTB- drug sensitive TB, DRTB- drug resistance TB, HIV- Human immunodeficiency virus, DM- Diabetes mellitus, HWC- Health and wellness centre, PHC- Primary health centre, CHC- Community health centre, DH- District hospital, PR- prevalence ratio, aPR- adjusted prevalence ratio.

*Residency- based on the districts in which the current residence lies.

**Private- all non-government health facilities.

***missing- data not available and entered.

ᶜcolumn percentage.

ʳrow percentage.

PLOS Global Public Health

Our data shows that only three in four households of index cases underwent contact investigation. TPT initiation was also lower than the national targets for all age groups. Children <5 years who have been a priority age group to focus on for contact investigation and TPT initiation also do not meet the program expectation of 90% TPT initiation [5].

Most of the predictors of not undergoing HHC investigation are health system-related factors. Those with current facility of care in tertiary care PHI and the private sector were less likely to undergo contact investigation. Those diagnosed by methods other than microscopy (available in primary care PHI) were also less likely to undergo contact investigation. This may be due to the non-implementation of home visits for contact investigation among those diagnosed at tertiary health centres, which are often located at a greater distance from home compared to primary healthcare facilities. Lack of communication between the tertiary care centres and private health care providers with the functionaries at PHC responsible for HHC investigation, inadequate training for primary health care workers, poor data quality, or a combination of the above may have contributed to HHC investigations not being done. These findings highlight the need for a systematic, patient-centred approach to decentralized primary care to be used for HHC investigation [21]. Studies conducted in similar settings in Africa and Asia using a decentralized approach have reported improved outcomes with respect to HHC investigation [12,13,22]. Studies from India have also shown similar results, where training of primary care staff and home visits were used [23,24].

The contacts of those with drug-resistant TB did not undergo HHC investigation. Drug-resistant TB is identified and managed at the tertiary care level using culture and LPA. TPT among HHCs of people with drug-resistant TB had been recently introduced and was being implemented phase-wise in all age groups [5]. This may have contributed to the lack of clarity during the study period. These two factors might have contributed to the HHC investigation not being done for people with drug-resistant TB.

The difference in HHC investigation between tribal and non-tribal districts may suggest a greater focus on the tribal population in the program in recent years due to a higher prevalence of TB in these communities [2,16]. Financial incentives for transport allowance in tribal TB units may also be contributing to better HHC investigation [18]. This means a need for greater emphasis on all districts in the state. A lesser proportion of women with TB disease undergoing HHC investigation may be because of unequal access to healthcare services in general, lower prevalence of TB among women, and lower awareness about the disease [16,25].

Incomplete individual-level data for those eligible for TPT resulted in an inability to understand contact-related factors in undergoing HHC investigation and TPT completion. The individual-level record of people identified with TB through contact investigation was also not accessible. This may be because of the recent prioritization of TB prevention in the program [5,26]. Missing data for screening for diabetes and HIV was also significantly associated with the HHC investigation not being done. This could be due to the data not being entered or screening not being done. Issues with the quality of data in *Ni-kshay* have been highlighted in other studies [27,28].

Monitoring of the program with respect to HHC investigation needs to be prioritized within routine program settings. This requires the introduction of selected indicators in the routine monthly and quarterly performance review reports (see **Table 2**). A summary of findings has been shared with the state officials. A formal meeting is planned with them to discuss the capacity-building initiative and the systematic use of a patient-centred approach in decentralized primary care for HHC investigation.

Future research should focus on reasons for the low yield of HHC investigation, the time delay in HHC investigation, and TPT completion rates.

## Conclusion

The study assessed the status of HHC investigation using program data for the state of Chhattisgarh and the extent of those diagnosed with TB, put on TPT, and factors predicting HHC. This study highlights proximal-level gaps in TB diagnosis and prevention among HHCs. The extent of HHC investigation, those diagnosed with TB, and those put on TPT was

**Table 2. Quarterly performance review indicators slide for the state program to review HHC investigation.**

| Indicators to be tracked | All age groups | Age < 5 years |
|---|---|---|
| No. of HHC investigations done/No. of pulmonary TB notified | | |
| Total number of HHCs identified | | |
| No. of people with TB detected through contact tracing/Total HHCs identified | | |
| No. of TPT initiated/HHCs identified | | |

TB- tuberculosis, TPT- TB preventive treatment.

suboptimal (for all ages and for children <5 years). Women, those from non-tribal districts, those diagnosed in PHI other than primary care level and in private healthcare facilities, diagnosed with tests other than microscopy, and missing data for HIV and diabetes screening were predictors of not undergoing HHC investigation. The findings suggest the need for decentralization of care and improvement in data quality. This information will be used to implement a capacity-building initiative in the state to improve and monitor the progress in HHC investigation.

## Supporting information

**S1 File. Annexure- Codebook used for data.**
(XLSX)

## Acknowledgments

This operational/ implementation research that resulted in this manuscript was conducted through the Structured Operational Research and Training Initiative (SORT IT), a global partnership led by the Special Program for Research and Training in Tropical Diseases at the World Health Organization (WHO/TDR). The model is based on a course developed jointly by the International Union Against TB and Lung Disease (The Union) and Medécins sans Frontières (MSF/Doctors Without Borders). This specific SORT IT course which resulted in this publication was part of year one of ICMR-National Institute of Epidemiology (ICMR-NIE) led TB SORT IT course 2024–26, with support and guidance from India's Central TB Division and WHO India. It was jointly developed and implemented by: ICMR-National Institute of Epidemiology (ICMR-NIE), Chennai, India; ICMR-National Institute for Research in TB (ICMR-NIRT), Chennai, India; Post Graduate Institute of Medical Education and Research (PGIMER), Chandigarh, India; FIND, New Delhi, India; Baroda Medical College, Vadodara, India; Narotam Sekhsaria Foundation, Mumbai, India; Government Medical College, Shahdol, India; All India Institute of Medical Sciences (AIIMS), Madurai, India; All India Institute of Medical Sciences (AIIMS), Bhathinda, India; Yenepoya Medical College, Mangaluru, India; and GMERS Gotri Medical College, Vadodara, India.

CM would also like to acknowledge his family, peers, mentors and co-workers from 'Sangwari- People's Association for Equity and Health', a not -for-profit health organization working in Surguja, Chhattisgarh and officials in the state TB office, Department of Health, the Government of Chhattisgarh for their support.

## Author contributions

**Conceptualization:** Chetanya Malik, Vishnu Gupta, Kalpita Shringarpure, Himanshu Abhay Gupte, Hemant Deepak Shewade, Vikash Ranjan Keshri, Narayan Tripathi, Khemraj Sonwani, Yogeshwar Kalkonde, Yogesh Jain.

**Data curation:** Chetanya Malik, Vishnu Gupta, Hemant Deepak Shewade.

**Formal analysis:** Chetanya Malik, Vishnu Gupta, Kalpita Shringarpure, Himanshu Abhay Gupte, Hemant Deepak Shewade.

**Investigation:** Chetanya Malik, Vishnu Gupta, Kalpita Shringarpure, Himanshu Abhay Gupte.

**Methodology:** Chetanya Malik, Vishnu Gupta, Kalpita Shringarpure, Himanshu Abhay Gupte, Hemant Deepak Shewade.

**Project administration:** Khemraj Sonwani.

**Supervision:** Kalpita Shringarpure, Himanshu Abhay Gupte, Hemant Deepak Shewade, Yogesh Jain.

**Visualization:** Chetanya Malik, Kalpita Shringarpure, Himanshu Abhay Gupte, Hemant Deepak Shewade, Vikash Ranjan Keshri, Narayan Tripathi, Khemraj Sonwani, Yogeshwar Kalkonde, Yogesh Jain.

**Writing – original draft:** Chetanya Malik, Vishnu Gupta, Kalpita Shringarpure, Himanshu Abhay Gupte, Hemant Deepak Shewade.

**Writing – review & editing:** Chetanya Malik, Vishnu Gupta, Kalpita Shringarpure, Himanshu Abhay Gupte, Hemant Deepak Shewade, Vikash Ranjan Keshri, Narayan Tripathi, Khemraj Sonwani, Yogeshwar Kalkonde, Yogesh Jain.

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
