## [Decision Letter · Decision Letter 0]

29 May 2025

PGPH-D-25-00886

Is household contact investigation a missing link for tuberculosis care in Chhattisgarh, India? – Operational research using programmatic data

Dear Dr. Malik,

Thank you for submitting your manuscript to PLOS Global Public Health. After careful consideration, we feel that it has merit but does not fully meet PLOS Global Public Health’s publication criteria as it currently stands. Therefore, we invite you to submit a revised version of the manuscript that addresses the points raised during the review process.

Please note that we have only been able to secure a single reviewer to assess your manuscript. We are issuing a decision on your manuscript at this point to prevent further delays in the evaluation of your manuscript. Please be aware that the editor who handles your revised manuscript might find it necessary to invite additional reviewers to assess this work once the revised manuscript is submitted. However, we will aim to proceed on the basis of this single review if possible. Please find their comments are available below.

The reviewers have raised a number of concerns that need attention, could you please revise the manuscript to carefully address the concerns raised?

We look forward to receiving your revised manuscript.

Kind regards,

Johanna Pruller, Ph.D.

PLOS Staff Editor

Journal Requirements:

Additional Editor Comments (if provided):

Reviewers' comments:

Reviewer's Responses to Questions

**Comments to the Author**

1. Does this manuscript meet PLOS Global Public Health’s publication criteria ? Is the manuscript technically sound, and do the data support the conclusions? The manuscript must describe methodologically and ethically rigorous research with conclusions that are appropriately drawn based on the data presented.

Reviewer #1: Yes

2. Has the statistical analysis been performed appropriately and rigorously?

Reviewer #1: Yes

3. Have the authors made all data underlying the findings in their manuscript fully available (please refer to the Data Availability Statement at the start of the manuscript PDF file)?

Reviewer #1: Yes

4. Is the manuscript presented in an intelligible fashion and written in standard English?

Reviewer #1: Yes

5. Review Comments to the Author

Reviewer #1: This study aims to assess the coverage of HHC investigation, proportions identified with TB and put on TPT for all age groups and age<5 years, and identified predictors of HHC investigation not done among those notified for bacteriologically confirmed pulmonary TB in Chhattisgarh, India. This is a good study, however, the authors may need to clarify some issues as mentioned below:

1. In the results, “of the 4221 households having an adult with bacteriologically confirmed pulmonary TB (index) between October to December 2023, contact investigation was conducted for 75% (n=3177). Aggregate contact tracing register data showed a total of 11670 contacts being screened and of them, TB was diagnosed in 0.9% (n=109) and TPT initiated for 66% (n=7740) contacts”.

The study aims to assess the coverage of HHC investigation. However, for the scope of the study, it could only report the proportion of households having bacteriologically confirmed pulmonary TB have contact investigation done (75), not report the proportion of number of HHC (11670 contacts) being screened among total number of HHC (of 4221 PTB index cases); and also the proportion of number of HHC (11670) being screened among total number of HHC of 3177 index cases (we could not sure that 100% of HHC of 3177 index cases being screened). This is need to be discussed in the discussion and mention in the limitation if could not provide this information.

2. The study result shows TB was diagnosed in 0.9% (n=109) among HHC being screened. This is high proportion, meaning prevalence of 934 per 100,000 (compare to 129 per 100,000 PTB notification; 373 per 100,000 TB prevalence in this region). However, line 49, “low yield of TB”, please clarify.

3. The study aims to assess the coverage of HHC investigation, proportions identified with TB and put on TPT for all age groups and age<5 years. As the results, of 11670 contacts screened, TB was diagnosed in 0.9% (n=109) and TPT initiated for 66% (n=7740) contacts. Of the identified contacts, 1230 were below the age of 5 years; of whom nine (0.7%) were diagnosed with TB and 73% (n=903) were initiated on TPT. May more detail analysis and present in another table for the comparison of proportion of identified TB in different groups.

4. Table 1, the percentage are row percent, but the clinical characteristic group the percentage are column percent (i.e. new (71.6%); resistant (7.5%), please check and correct.

5. Of 78 HHC of resistant cases, all 78 (100%) was not being screened for TB. This should be highlight and discuss for improvement. (This proportion was present as column percent in table 2, should be row percent as mentioned above).

6. PLOS authors have the option to publish the peer review history of their article (what does this mean? ). If published, this will include your full peer review and any attached files.

**Do you want your identity to be public for this peer review?** For information about this choice, including consent withdrawal, please see our Privacy Policy .

Reviewer #1: No

---

## [Decision Letter · Decision Letter 1]

17 Jul 2025

PGPH-D-25-00886R1

Is household contact investigation a missing link for tuberculosis care in Chhattisgarh, India? – Operational research using programmatic data

Dear Dr. Malik,

Thank you for revising your manuscript as per the comments of the reviewers. After careful consideration, we feel that the modifications are adequate. However, there are a few more concerns raised by one of the reviewers. Therefore, we invite you to submit a revised version of the manuscript that addresses these points raised during the review process.

Query of the reviewer on the choice of Poisson regression for the multivariable analysis needs to be explained. A few more suggested changes in the language and discussion will improve the quality of the manuscript. The term contact investigation may be retained and not changed to 'contact tracing' as suggested.

We look forward to receiving your revised manuscript.

Kind regards,

Sonali Sarkar

Academic Editor

Journal Requirements:

Reviewers' comments:

Reviewer's Responses to Questions

**Comments to the Author**

1. If the authors have adequately addressed your comments raised in a previous round of review and you feel that this manuscript is now acceptable for publication, you may indicate that here to bypass the “Comments to the Author” section, enter your conflict of interest statement in the “Confidential to Editor” section, and submit your "Accept" recommendation.

Reviewer #1: All comments have been addressed

Reviewer #2: All comments have been addressed

Reviewer #3: All comments have been addressed

2. Does this manuscript meet PLOS Global Public Health’s publication criteria ? Is the manuscript technically sound, and do the data support the conclusions? The manuscript must describe methodologically and ethically rigorous research with conclusions that are appropriately drawn based on the data presented.

Reviewer #1: Yes

Reviewer #2: Yes

Reviewer #3: Yes

3. Has the statistical analysis been performed appropriately and rigorously?

Reviewer #1: Yes

Reviewer #2: Yes

Reviewer #3: Yes

4. Have the authors made all data underlying the findings in their manuscript fully available (please refer to the Data Availability Statement at the start of the manuscript PDF file)?

Reviewer #1: Yes

Reviewer #2: Yes

Reviewer #3: Yes

5. Is the manuscript presented in an intelligible fashion and written in standard English?

Reviewer #1: Yes

Reviewer #2: Yes

Reviewer #3: Yes

6. Review Comments to the Author

Reviewer #1: I have no further comments

Reviewer #2: (No Response)

Reviewer #3: 1. The outcome variable - of HHC investigation not done is a dichotomous variable. Why is Poisson regression model used instead of logistic model.

2. Line no. 40-41, please indicate number and percentage

3. Line no. 172, Is contact tracing a better word?

4. Lines no. 228 - 233 - Unclear what authors are trying to convey. Please rephrase

5. Line no. 240 - Please mention to which geographic context does this meta-analysis refer to?

6. Line no. 262-263 - How is this point substantiating lack of follow-up among patients diagnosed in secondary / tertiary facilities?

7. PLOS authors have the option to publish the peer review history of their article (what does this mean? ). If published, this will include your full peer review and any attached files.

**Do you want your identity to be public for this peer review?** For information about this choice, including consent withdrawal, please see our Privacy Policy .

Reviewer #1: No

Reviewer #2: No

Reviewer #3: No

---

## [Editor Report · Decision Letter 2]

19 Aug 2025

PGPH-D-25-00886R2

Is household contact investigation a missing link for tuberculosis care in Chhattisgarh, India? – Operational research using programmatic data

Dear Dr. Malik,

Thank you for submitting your manuscript to PLOS Global Public Health. After careful consideration, we feel that it has merit but does not fully meet PLOS Global Public Health’s publication criteria as it currently stands. Therefore, we invite you to submit a revised version of the manuscript that addresses the points raised during the review process.

Language needs to be checked for repetitiveness.   

We look forward to receiving your revised manuscript.

Kind regards,

Sonali Sarkar

Academic Editor

Journal Requirements:

Additional Editor Comments:

Thank you for modifying the manuscript as per the suggestions of the reviewers. However, a thorough language check will improve the quality of the manuscript. Please modify the following portions.

1. Lines 230-233, where an explanation has been provided for the discrepancies in the numbers of the HHCs investigated. There is repetition of information in the two sentences.

2. Lines 261-266 need to be simplified to convey the message clearly. You may consider the following to replace these lines.

"Lack of communication between the tertiary care centres and private health care providers with the functionaries at PHC responsible for HHC investigation, inadequate training for primary health care workers, poor data quality or a combination of the above may have contributed to HHC investigations not being done."

Overall, the number of words and sentences can be reduced to make the manuscript succinct.
---

## [Editor Report · Decision Letter 3]

10 Sep 2025

Is household contact investigation a missing link for tuberculosis care in Chhattisgarh, India? – Operational research using programmatic data

PGPH-D-25-00886R3

Dear Dr. Malik,

We are pleased to inform you that your manuscript 'Is household contact investigation a missing link for tuberculosis care in Chhattisgarh, India? – Operational research using programmatic data' has been provisionally accepted for publication in PLOS Global Public Health.

Best regards,

Sonali Sarkar

Academic Editor